# Multi-Objective Optimized Fuzzy Clustering for Detecting Cell Clusters from Single-Cell Expression Profiles

**DOI:** 10.3390/genes10080611

**Published:** 2019-08-13

**Authors:** Saurav Mallik, Zhongming Zhao

**Affiliations:** 1Center for Precision Health, School of Biomedical Informatics, The University of Texas Health Science Center at Houston, Houston, TX 77030, USA; 2Department of Biomedical Informatics, Vanderbilt University Medical Center, Nashville, TN 37203, USA

**Keywords:** cluster validity indices, fuzzy clustering, Limma, multi-objective optimization, single cell sequencing, TOPSIS

## Abstract

Rapid advance in single-cell RNA sequencing (scRNA-seq) allows measurement of the expression of genes at single-cell resolution in complex disease or tissue. While many methods have been developed to detect cell clusters from the scRNA-seq data, this task currently remains a main challenge. We proposed a multi-objective optimization-based fuzzy clustering approach for detecting cell clusters from scRNA-seq data. First, we conducted initial filtering and SCnorm normalization. We considered various case studies by selecting different cluster numbers (cl = 2 to a user-defined number), and applied fuzzy c-means clustering algorithm individually. From each case, we evaluated the scores of four cluster validity index measures, Partition Entropy (PE), Partition Coefficient (PC), Modified Partition Coefficient (MPC), and Fuzzy Silhouette Index (FSI). Next, we set the first measure as minimization objective (↓) and the remaining three as maximization objectives (↑), and then applied a multi-objective decision-making technique, TOPSIS, to identify the best optimal solution. The best optimal solution (case study) that had the highest TOPSIS score was selected as the final optimal clustering. Finally, we obtained differentially expressed genes (DEGs) using Limma through the comparison of expression of the samples between each resultant cluster and the remaining clusters. We applied our approach to a scRNA-seq dataset for the rare intestinal cell type in mice [GEO ID: GSE62270, 23,630 features (genes) and 288 cells]. The optimal cluster result (TOPSIS optimal score= 0.858) comprised two clusters, one with 115 cells and the other 91 cells. The evaluated scores of the four cluster validity indices, FSI, PE, PC, and MPC for the optimized fuzzy clustering were 0.482, 0.578, 0.607, and 0.215, respectively. The Limma analysis identified 1240 DEGs (cluster 1 vs. cluster 2). The top ten gene markers were *Rps21, Slc5a1, Crip1, Rpl15, Rpl3, Rpl27a, Khk, Rps3a1, Aldob* and *Rps17*. In this list, *Khk* (encoding ketohexokinase) is a novel marker for the rare intestinal cell type. In summary, this method is useful to detect cell clusters from scRNA-seq data.

## 1. Introduction

Rapid technology development in sequencing over the last two decades has made the transcriptomic analysis of cells and tissues more reliable and informative [1]. Cells are basic units of organisms and the building blocks of various complex tissues; they are controlled by many factors that affect their cell status and features (e.g., cell type specific expression, senescence). Quantification of the mRNA transcripts in genome-wide basis is useful to characterize the molecular circuitries as well as cellular states. In general, such datasets are accumulated with higher spatial resolution, whereas the single-cell RNA sequencing (scRNA-seq) permits to conduct the transcriptome-wide analyses of single cells to discover the interesting biomedical insights as well as biological perception [1,2]. As a heterogeneous cell population, scRNA-seq shows the levels of gene expression for each individual cell, while in bulk-tissue RNA sequencing, mean value of the expression signature in the basis of their cell population level has been evaluated. ScRNA-seq needs the isolation as well as lysis of the single cells, the transformation of their corresponding RNA to cDNA, and the amplification of the cDNA to produce the high-throughput sequencing libraries.

Here one important factor is evaluation of sensitivity of the scRNA-seq technique i.e., the probability for capturing and transforming a corresponding mRNA transcript that presents in a single cell to a cDNA biomolecule that present in the library. Another important factor is evaluation of the accuracy, i.e., how good the read quantification correlates with the actual expression of the mRNA. The third factor is the evaluation of precision in which the amplification happens, i.e., the technical variation related to the quantification. Moreover, the integration of precision, sensitivity, and the number of cells that have been used in analysis, evaluates the power for identifying the relative differences in the levels of expression. To choose a proper and efficient method among all the state-of-the-art scRNA-seq techniques, it requires to measure those parameters accurately. Major advantages and limitations of various scRNA-seq techniques have been well discussed in the field. In addition, Smart-seq technique was optimized for the full-length coverage, sensitivity, cost as well as accuracy [3]. Next, the enhanced version of this technique (enhanced Smart-seq2 technique) was developed by Picelli et al. in 2014 [4] that is also useful in various works [5,6,7].

It is well-known that the characterization of entire cell types in any complex tissue needs the processing of at least a couple of thousand single cells [8]. Larger sample size would make the evaluation better. It is true that a larger number of cell type-related transcripts are not identified in current scRNA-seq because of the failure during the stage of amplification and the relative limitation of short read coverage. Consequently, a limited number of cell type-related genes might fail to affect the downstream analysis regime in sufficient way. Current discovery of droplet-based single-cell transcriptomics is helpful to perform the parallel profiling of the tens of thousands of the single cells at significantly low expense per cell. Various studies within the range of the transcriptomes of cells from 20 k and 70 k have already published [9,10,11].

One of the major challenges of single-cell transcriptomics is the identification of the clusters of cells. Most recently, Kiselev et al. provided a comprehensive review of use and challenges of different unsupervised clustering algorithms on scRNA-seq data [12]. Unsupervised learning (clustering) has a key role to analyze scRNA-seq data since it can be applied to detect the putative cell types. In that review article, the advantages and shortcomings associated with the biological interpretation as well as annotation of the evolved clusters in different clustering algorithms were described. Andrews and Hemberg provided a survey of various computational methods to determine cell populations for scRNA-seq data [13]. Furthermore, Zhu et al. developed a novel technique, semisoft clustering which could classify both the pure cell types from the individual cells of the scRNA-seq gene expression profile [14]. Diaz-Mejia et al. compared four benchmarked algorithms, GSVA, CIBERSORT, ORA and GSEA to assign the cell type labels to the cell clusters in the scRNA-seq data [15]. Diaz-Mejia et al. chose the scRNA-seq datasets from various sources such as peripheral blood mononuclear cells, liver as well as retinal neurons for which the corresponding reference cell type-based gene expression signatures would be available. In addition, rare cell populations also play a significant role to detect the pathogenesis of cancer mediating angiogenesis, immune responses in cancer as well as other diseases. Antigen-related T cells are mandatory for forming the immunological memory [16,17,18]. Endothelial progenitor cells are potential biomarkers for the tumor angiogenesis [19,20]. Circulating tumor cells and their usability in management of the cancer as well as clinical data study were represented in Krebs et al. [21]. Stem cells can replace the damaged cells, and can also make treatment on several diseases such as heart diseases, Parkinson’s disease, etc. [22]. In general, the number of methods developed for identifying the rare cell transcriptomes is very few. Among them, two algorithms (RaceID [23] and GiniClust [24]) work nicely. The two limitations of these algorithms are high elapsed time and lack of memory efficiency in case of large-scale oversized scRNA-seq profile. Hence, there are many approaches that were developed for satisfying different individual objectives for the analysis of the single-cell sequencing data.

In the literature, there are many articles about the identification of gene signature [25,26,27,28] or, biomarker [29,30,31] or, gene module [26,32,33] or, transcriptome analysis [34,35] for microarray/RNA-seq data and the integration of multi-omics data [36,37,38,39,40]. Among them, there are a few papers using fuzzy clustering for the microarray/RNA-seq data [41,42]. However, there has been no study yet using multi-objective optimization and fuzzy clustering together on single-cell RNA-seq data. Therefore, in this article, we developed a new computational framework to identify cell clusters using multi-objective optimization and fuzzy clustering together on scRNA-seq data. Our method has several advantages in the application of fuzzy clustering and multi-objective optimization strategy. The intuition behind using fuzzy clustering and TOPSIS (“Technique for Order Preference by Similarity to an Ideal Solution”) multi-objective optimization strategy is described as follows: (i) Fuzzy C-Means (FCM) is a kind of clustering strategy in which each sample point belonging to the cluster is characterized by its membership function. In general, FCM tries to maintain the membership matrix of the input dataset that has been updated on every iteration by estimating the associated weight of every sample point to evaluate its degree of membership. The summation of every sample point towards all the clusters is unity. The main benefits of this strategy to scRNA-seq data include its capability to form clusters of the overlapped sample points and the results satisfy the property of convergence. The potential limitations of the cluster validity are that the prior necessity of c value is required for the quality clustering outcomes and outliers might be assigned to the similar membership value in every cluster. These limitations make it less desirable for using any kind of gene expression data. (ii) TOPSIS method is used to identify the set of multi-objective optimized clusters. In the scRNA-seq data, the number of cell clusters varies among data sets. In this study, we attempted to identify the best set of multi-objective optimized clusters as measured by the quality of clustering, i.e., different clustering validity index measures. (iii) We performed a comparative analysis of our proposed method with the existing k-means clustering method. The comparative analysis indicated that our method outperformed over k-means clustering method.

Specifically, cell filtering and gene filtering were first performed for the single-cell data, and then normalized the data using the robust SCnorm normalization, respectively. We then considered various case studies through the selection of different cluster sizes, and used fuzzy c-means clustering algorithm, individually. For each case, we obtained the four cluster validity index measures, Partition Entropy, Partition Coefficient, Modified Partition Coefficient, and Fuzzy Silhouette Index. We set these four measures as objective functions in which first index was treated as minimization objective and rest of these three were assumed to be maximization objectives. Next we applied a multi-objective decision-making technique, TOPSIS with providing equal preference to each objective for detecting the multi-objective optimal (best) solution. From the optimal solution, we obtained the corresponding optimal cluster size along with cell-cluster information. We then performed Limma statistical package using the cell-cluster information to identify differentially expressed genes for each evolved cluster while compared to the rest. Furthermore, the top ten differentially expressed genes were considered to be the potential gene markers for the cluster. Furthermore, we conducted the KEGG pathway and Gene Ontology analyses through DAVID online database. Finally, our framework provided the multi-objective optimized clusters as well as potential gene markers for each cluster that might be useful to any scRNA-seq data.

## 2. Materials and Methods

In this article, we provided an extensive analysis to identify the single-cell clusters and gene markers, respectively using multi-objective optimization-based fuzzy clustering for a scRNA-seq gene expression dataset. See Figure 1 for the flowchart.

### 2.1. Initial Filtering

First, we performed some filtering analysis on the initial data. According to the latest literature search [23,43], it had been noticed that the filtering criteria was not fixed anymore. Hence, we chose the standard cutoffs for cell filtering as well as gene filtering. However, first we transformed the count data to a Boolean matrix where the non-zero values were replaced by 1. Next, we counted the summation of all the non-zero values for each cell, and then chose those cells which contained the summation score greater than 2000. After that, we focused on gene filtering. Here we only selected those genes (features) that contained the summation of all the corresponding non-zero values greater than 3 and less than 500. Finally, we computed the feature-wise variance for each gene, and then chose some top highest variant genes for the next step. Thereafter, we identified the count-depth relationship through plot and then applied a recent robust normalization technique, SCnorm [44] made for single-cell sequencing data.

Of note, prior to normalize through SCnorm normalization [44], the relationship between the expression counts and the corresponding sequencing depth (named as the count-depth relationship or slope) for the experimental data should be verified. SCnorm normalizes across the cells for eliminating the effect of the sequencing depth on the counts. The genes were initially partitioned into 10 equally sized groups depending upon their non-zero median expression. In SCnorm, only those genes that had at least 10 non-zero expression values, were selected be default. SCnorm had initiated at the value of the parameter K equal to 1 that helped to normalize the data with the assumption that all the genes had to be normalized in a single group. The sufficiency of the score K = 1 was estimated through determining the normalized count-depth relationship. To do so, all the genes were divided into 10 groups depending on their corresponding non-zero unnormalized median expression scores (considering equal group-size) and evaluated the mode for each corresponding group. Whenever all 10 modes were within 0.1 of zero, the value K = 1 would be sufficient. While any of those modes was less than −0.1 or greater than 0.1, the SCnorm method attempted to normalize considering K = 2 and repeated the group-wise normalizations along with the corresponding estimation. It would continue until all those modes were within 0.1 of zero. Moreover, SCnorm method initially tried to fit the corresponding model for the value K = 1, and then subsequently increases the value of K until an approximate satisfactory stopping point had been reached. As the gene expression generally increased proportionally while increasing the sequencing depth, the count-depth relationships were required to evaluate near 1 for all the genes. In general, for single-cell data, those relationships were somehow variable across the genes. However, after the SCnorm normalization, the count-depth relationship could be evaluated on the normalized data profile where the slopes near zero signified successful normalization.

### 2.2. Fuzzy Clustering for Finding Optimized Cell Clusters

After normalization, we applied a well-known clustering algorithm, fuzzy c-means clustering for the initial number of clusters, cl=2,3,…udc (where udc be a user-defined cluster size), signifying (udc-1) number of case studies, individually, and then computed four cluster validity indices: Partition Entropy, Partition Coefficient, Modified Partition Coefficient, and Fuzzy Silhouette Index from each case study.

Fuzzy c-means [45] is a clustering algorithm based upon fuzzy membership concept in the domain of machine learning. Let Fuzzy c-means clustering algorithm makes partitions of *n* data features (data points) $X={\left\{x_{1},x_{2},x_{3},\ldots ,x_{n}\right\}}_{n\times a}$ into cl (1≤cl≤n) fuzzy clusters while every feature contains *a* number of attributes. Suppose $Cen={\left\{cen_{1},cen_{2},cen_{3},\ldots ,cen_{cl}\right\}}_{cl\times a}$ stands for the set of cluster centers, and $U={\left[u_{jq}\right]}_{cl\times n}$ is cl×n matrix of the membership degrees where $u_{jq}$ denotes the membership degree of $q^{th}$ feature to $cl^{th}$ cluster center. The above matrix satisfies the following criteria: $\sum_{j=1}^{cl}u_{jq}=1$, $u_{jq}\geq 0$ and $u_{jq}\in \left[0,1\right]$.

The fuzzy c-means algorithm applies the following objective function to obtain the optimal solution of the corresponding fuzzy optimal clustering. The objective function $J_{fcm}$ is be defined in the following:(1)$$J_{fcm}=\sum_{j=1}^{cl}\sum_{q=1}^{n}u_{jq}^{m}\left|\right|x_{q}-cen_{j}{\left|\right|}^{2},$$
where *m* (1≤m≤∞) is the fuzzification coefficient denoting the fuzzy degree of clustering. In our work, we used m = 1. Here ||∗|| be any norm measuring the similarity between the cluster center and any measured data. The objective function $J_{fcm}$ should be minimized.

The minimization of the objective function is performed through Lagrange multiplier technique under the constraint $\sum_{j=1}^{cl}u_{jq}=1$ (q=1,2,3,…,n), whereas the membership degree and the cluster centers have been updated by the following equations:(2)$$u_{jq}=\sum_{s=1}^{cl}{\left(\frac{\left|\right|x_{q}-cen_{j}\left|\right|}{\left|\right|x_{q}-cen_{s}\left|\right|}\right)}^{2/(m-1)},$$
(3)$$cen_{j}=\frac{\sum_{q=1}^{n}\left(u_{jq}^{m}x_{q}\right)}{\sum_{q=1}^{n}u_{jq}^{m}}.$$

The algorithm terminates while the criteria $max_{jq}|u_{jq}^{i+1}-u_{jq}^{i})|\leq \epsilon$ is satisfied, whereas ϵ be a termination constant that lies between 0 and 1, while *i* denotes the iteration step id. This algorithm converges to either a local minimum or saddle point of the objective function $J_{fcm}$.

### 2.3. Measuring Cluster Validity Index Measures

Partition Entropy (PE) [46,47] and Partition Coefficient (PC) [47,48] are two cluster validity indices that were developed by James C. Bezdek. PE and PC were defined as follows:(4)$$PE=-\frac{i}{n}\sum_{j=1}^{cl}\sum_{q=1}^{n}u_{jq}*log_{e}u_{jq},$$
and
(5)$$PC=\frac{i}{n}\sum_{j=1}^{cl}\sum_{q=1}^{n}u_{jq}^{2},$$

Modified Partition Coefficient (MPC) [49,50] was introduced for correcting the monotonic trend of PC. The values of MPC lie between 0 and 1. MPC was defined as follows:(6)$$MPC=1-\frac{cl}{cl-1}\left(1-PC\right),$$

Fuzzy Silhouette Index (FSI) [50,51] is such a measure where it selects the two such clusters in which $x_{q}$ contains the highest membership degrees. FSI was described in the following:(7)$$FSI=\frac{\sum_{q=1}^{n}\left(u_{1q}-u_{2q}\right)S\left(x_{q}\right)}{\sum_{q=1}^{n}\left(u_{1q}-u_{2q}\right)},$$
where
(8)$$S\left(x_{q}\right)=\frac{\beta \left(x_{q},gt_{j}\right)-\delta \left(x_{q},gt_{j}\right)}{max\{\beta \left(x_{q},gt_{j}\right),\delta \left(x_{q},gt_{j}\right)\}}$$

Here, a data feature (point) $x_{q}$ is in the part of the cluster $gt_{j}$, ($gt_{j}\in \left(gt_{1},gt_{2},gt_{3},\ldots ,gt_{cl}\right)$); whereas $\delta (x_{q},gt_{j})$ is basically the intra-cluster distance that signifies the mean distance between $x_{q}$ and all other points that belong to the same cluster, $gt_{j}$. On the other hand, $\beta (x_{q},gt_{j})$ is an inter-cluster distance that denotes the distance between $x_{q}$ and its neighbor cluster closest to the cluster $gt_{j}$.

For obtaining the optimal clusters, PC, MPC, and FSI should be maximized, while PE should be minimized.

### 2.4. Identifying Optimal Fuzzy Clustering Solution Using Multi-Objective Decision-Making Model

After producing the values of these four measures for each case study, we applied a multi-criterion decision-making model, TOPSIS (“Technique for Order Preference by Similarity to an Ideal Solution”) [52,53].

First the decision matrix was constructed, and the weight of criteria was evaluated. Suppose, $Y=y_{ij}$ is a decision matrix and $Wt=(wt_{1},wt_{2},wt_{3},\ldots ,wt_{n})$ is a weight vector, ($y_{ij}\in ℜ$, $wt_{j}\in ℜ$ and $wt_{1}+wt_{2}+wt_{3}+\ldots +wt_{n}=1$). Of note, the criteria of the functions could be one of the two types: (i) benefit functions (maximization is better), or, (ii) cost functions (minimization is better).

Next, the normalized decision matrix was computed. In this step, different attribute dimensions were converted into the non-dimensional attributes that allowed the comparisons toward the criteria. Since the different criteria were generally evaluated in different units, the evaluation scores in the corresponding matrix *Y* were required to convert into a normalized scale. The normalized values could be performed through one of the popular standardized formulas. The most useful techniques of computing the normalized score $n_{ij}$ were as follows.
(9)$$n_{ij}=\frac{y_{ij}}{\sqrt{\sum_{i=1}^{m}y_{ij}^{2}}},$$
(10)$$n_{ij}=\frac{y_{ij}}{\max_{i}y_{ij}},$$
(11)$$n_{ij}=\left\{\begin{array}{cc} \frac{y_{ij}-\min_{i}y_{ij}}{\max_{i}y_{ij}-\min_{i}y_{ij}}, & \mathrm{if}C_{i}\mathrm{denotes}\mathrm{benefit}\mathrm{objective}\left(\mathrm{criteria}\right)\left(↑\right)\\ \frac{\max_{i}y_{ij}-y_{ij}}{\max_{i}y_{ij}-\min_{i}y_{ij}}, & \mathrm{if}C_{i}\mathrm{becomes}\mathrm{cost}\mathrm{objective}\left(↓\right) \end{array}\right.$$
for i=1,2,3,…,m, and j=1,2,3,…,n.

After that, the weighted normalized decision matrix $v_{ij}$ was determined through the following manner:(12)$$v_{ij}=wt_{j}n_{ij},$$
for i=1,2,3,…,m and j=1,2,3,…,n. where $wt_{j}$ denoted the corresponding weight of *j*-th criterion, and $\sum_{j=1}^{n}wt_{j}=1$.

Next, the positive ideal alternative and the negative ideal alternative were evaluated. The ideal positive solution was the solution which maximized the benefit portion of the criteria as well as minimized the cost portion of criteria. On the other hand, the negative ideal solution was the solution that maximized the cost section of the criteria along with minimizing the benefit section of the criteria. Positive ideal solution, $AB^{+}$ contained the following form:(13)$$AB^{+}=\left(v_{1}^{+},v_{2}^{+},v_{3}^{+},\ldots ,v_{n}^{+}\right)=\left(\left(\max_{i}v_{ij}|j\in I\right),\left(\min_{i}v_{ij}|j\in J\right)\right).$$
and negative ideal solution $AB^{-}$ contained as follows:(14)$$AB^{-}=\left(v_{1}^{-},v_{2}^{-},v_{3}^{-},\ldots ,v_{n}^{-}\right)=\left(\left(\min_{i}v_{ij}|j\in I\right),\left(\max_{i}v_{ij}|j\in J\right)\right),$$
where *I* and *J* were related to the benefit portion of the criteria and the cost portion of the criteria, respectively, i=1,2,3,…,m and j=1,2,3,…,n.

The separation measures were computed from the positive ideal solution and from the negative ideal solution. Here, in TOPSIS approach, distance metrics were used. The separation of every alternative obtained from the positive ideal solution was demonstrated as below:(15)$$ds_{i}^{+}={\left(\sum_{j=1}^{n}{\left(v_{ij}-v_{j}^{+}\right)}^{p}\right)}^{1/p},$$
where *i* = 1, 2, 3, …, *m*, and p≥1. On the other hand, the separation of every alternative obtained from the negative ideal solution was described in the following:(16)$$ds_{i}^{-}={\left(\sum_{j=1}^{n}{\left(v_{ij}-v_{j}^{-}\right)}^{p}\right)}^{1/p},$$
where *i* = 1, 2, 3, …, *m*, and p≥1.

Now in the case of p=2, the most useful traditional n-dimensional Euclidean metrics were produced as below:(17)$$ds_{i}^{+}=\sqrt{\sum_{j=1}^{n}{\left(v_{ij}-v_{j}^{+}\right)}^{2}},$$
and
(18)$$ds_{i}^{-}=\sqrt{\sum_{j=1}^{n}{\left(v_{ij}-v_{j}^{-}\right)}^{2}},$$
where *i* = 1, 2, 3, …, *m*.

After that, we computed the relative closeness (RC) of *i*-th alternative $AB_{j}$ with respect to $AB^{+}$ that was described as below:(19)$$RC_{i}=\frac{ds_{i}^{-}}{ds_{i}^{-}+ds_{i}^{+}},$$
where $0\leq RC_{i}\leq 1$, and *i* = 1, 2, 3, …, *m*. Finally, we ranked a set of alternatives in descending order of the corresponding score of $RC_{i}$.

In the TOPSIS algorithm, we provided the scores of the four cluster validity index measures for the (udc-1) number case studies (cl=2,3,…,udc), where Partition Coefficient, Modified Partition Coefficient, and Fuzzy Silhouette Index were three objectives to be maximized, and Partition Entropy was the objective to be minimized. Using this multi-objective optimization technique with providing equal weight to each objective, we determined the TOPSIS optimal score for each study. Higher score signified here better rank. The top ranked case study denoted here the final optimal solution. Hence, we only considered the final optimal solution among those case studies, and remaining solutions were discarded. The four cluster validity indices for the optimal solution that signified the quality of the underlying clustering, were represented in the final clustering result. From the optimal clustering, we also obtained the number of best optimal cluster size used for the clustering, and also cluster information of the cells that would be used for the next step, differentially expressed gene identification.

### 2.5. Identification of Differentially Expressed Genes through Statistical Test

In this step, we applied Limma statistical tool [40,54] that used Empirical Bayes test using the cluster information of the cells for each resultant cluster in compared to the rest of the clusters, and generated the *p*-value for each gene (feature). The *p*-values were then adjusted by the most stringent *p*-value correction strategy, Bonferroni method [55]. Those genes that contained adjusted *p*-value less than 0.05, were considered to be differentially expressed genes. The top genes were then searched in the literature to know whether there was any connection of them with any disease or biological function or not.

### 2.6. Gene Set Enrichment Analysis

We conducted KEGG pathway and Gene Ontology (GO) enrichment analysis for the differentially expressed genes for each resultant cluster vs. the rest clusters using DAVID online tool. After this analysis, we considered those pathways or GO-terms were significant if the Bonferroni corrected *p*-values were less than 0.05.

### 2.7. Identification of Novel Gene Markers

The top genes that were found to be associated with neither any disease or biological function in any literature, nor significantly enriched in any KEGG pathway or GO-term, were chosen as novel gene markers.

## 3. Results

### 3.1. Source Dataset

In this study, we first proposed single-cell cluster identification method based on multi-objective optimization for scRNA-seq gene expression data. The R code is available at https://github.com/sauravmtech2/MOO-FUZZ-for-scRNAseq-. Then, we evaluated the method by using the count data of “*Whole Organoid Replicate 1*” within a single-cell mRNA sequencing dataset (GEO ID: GSE62270). This dataset has initially contained 23,630 features (genes) and 288 cells in mice. Here, we provided our evaluate results.

### 3.2. Filtering Analysis

In the cell filtering, we selected a total of 206 cells in this step. In the gene filtering, the number of the selected filtered genes became 11,466. After computing the gene-wise variance, we chose top 3000 highest variant genes for the later step. We illustrated the count-depth relationship plot in Figure 2A. Next, we then normalized the data using the SCnorm normalization technique. In our study, the value of K in count-depth relationship plot was evaluated 4 (the four groups of slopes colored as sky, blue, magenta and red sequenced from lowest to highest expression median score) while all those 10 slope densities had the absolute slope mode < 0.1 (default cutoff). However, the normalization of the highly expressed genes would be good, whereas the lowly expressed (and moderately expressed) genes might be over-normalized and they generated negative slopes.

### 3.3. Fuzzy Clustering of Cells

For clustering the cells, we applied fuzzy c-means clustering for different initial number of clusters, cl=2,3,…,10 (nine case studies as udc = 10 here), individually, and produced the values of the four cluster validity indices: FSI, PE, PC, and MPC. See Table 1 for details.

### 3.4. Finding Optimal Clustering (Solution) Using Multi-Objective Optimization

We used TOPSIS multi-objective optimization technique on the nine case studies using FSI, PC, and MPC scores as maximal objectives and PE as minimal objective, and obtained the final optimal score of these nine case studies. The higher TOPSIS optimal score signified better rank. Here the case study for cl=2 produced highest TOPSIS optimal score (=0.8584), and hence, it had topmost optimal rank. Now we chose the result for this case study and the results of the remaining case studies were simply eliminated. See Figure 2A for the count-depth relation plot during the intermediate step, SCnorm normalization. We provided Table 1 for the details of the TOPSIS optimal scores and Table 2 for the TOPSIS optimal ranks. We obtained 115 cells in one cluster and remaining 91 cells in the other cluster. The participating cells of the evolved clusters were notified in Table 3. In addition, we included a boxplot representation of those membership values where their mean score for each cluster was clearly illustrated Figure 2B. The fuzzy membership degree values of these cells in each of these two resultant clusters are illustrated in Figure 2C. However, the graphical representation of the TOPSIS optimal scores for the nine case studies were represented in Figure 3A, whereas the overall representation of these two resultant clusters (PCA plot) were depicted in Figure 3B. The obtained values of these four cluster validity indices, FSI, PE, PC, and MPC for the optimized fuzzy clustering were 0.482, 0.578, 0.607 and 0.215, respectively. Of note, as shown in Figure 3A, the TOPSIS optimal score decreases every time when the number of clusters decreases. In addition, in the experimental data, there was small number of cells (only 206 cells) in the single organoid that we chose before clustering. Therefore, we checked the first nine cases (#cluster = 2 to 10) because those clusters had good scores to evaluate. The case study that had highest optimal score, was finally considered to be the final outcome of clustering, where other case studies were simply neglected. In addition, since the cell size in the data was small and the curve of optimal score was going down every time when increasing the cluster size, there was no need to check for the cases with cluster size >10.

### 3.5. Identification of Differentially Expressed Genes through Statistical Test

After obtaining the cluster information of the cells, we applied Limma tool (cluster 1 vs. cluster 2) and obtained a total of 1240 differentially expressed genes whose Bonferroni adjusted *p*-values were less than 0.05. The top ten genes (markers) were *Rps21* (adjusted *p*-value = $4.36\times 10^{-37}$), *Slc5a1* (adjusted *p*-value = $7.47\times 10^{-37}$), *Crip1* (adjusted *p*-value = $1.28\times 10^{-36}$), *Rpl15* (adjusted *p*-value = $5.71\times 10^{-35}$), *Rpl3* (adjusted *p*-value = $6.49\times 10^{-35}$), *Rpl27a* (adjusted *p*-value = $1.50\times 10^{-34}$), *Khk* (adjusted *p*-value = $4.09\times 10^{-34}$), *Rps3a1* (adjusted *p*-value = $1.00\times 10^{-33}$), *Aldob* (adjusted *p*-value = $1.19\times 10^{-33}$) and *Rps17* (adjusted *p*-value = $1.84\times 10^{-33}$). Furthermore, we provided a plot of rankwise adjusted *p*-value for the differentially expressed genes in Figure 4.

### 3.6. Gene Set Enrichment Analysis

Furthermore, we conducted gene set enrichment analysis using online tool DAVID and obtained significant KEGG pathway and Gene Ontology (GO) terms (three terms: Biological Process: BP, Cellular Component: CC and Molecular Function: MF). After Bonferroni correction, we had 18 significant KEGG pathways, 37 significant GO:BP terms, 55 significant GO:CC terms and 23 significant GO:MF terms. The top three significant KEGG pathways were Ribosome (Bonferroni corrected *p*-value = 1.08 $\times \;10^{-59}$ and 94 participating genes), Spliceosome (Bonferroni corrected *p*-value = 1.02 $\times \;10^{-19}$ and 54 participating genes), Biosynthesis of antibiotics (Bonferroni corrected *p*-value = 1.61 $\times \;10^{-8}$ and 52 participating genes). Details were summarized in Table 4.

The top five GO:BP terms were GO:0006412 translation (Bonferroni corrected *p*-value = 1.34 $\times \;10^{-70}$), GO:0008380 RNA splicing (Bonferroni corrected *p*-value = 9.38 $\times \;10^{-23}$), GO:0006397 mRNA processing (Bonferroni corrected *p*-value = 1.18 $\times \;10^{-20}$), GO:0055114 oxidation-reduction process (Bonferroni corrected *p*-value = 8.79 $\times \;10^{-14}$), GO:0006413 translational initiation (Bonferroni corrected *p*-value = 3.88 $\times \;10^{-12}$) (Table 5), where the top five GO:CC terms were GO:0030529 intracellular ribonucleoprotein complex (Bonferroni corrected *p*-value = 9.24 $\times \;10^{-97}$), GO:0070062 extracellular exosome (Bonferroni corrected *p*-value = 2.26 $\times \;10^{-78}$), GO:0005840 ribosome (Bonferroni corrected *p*-value = 2.70 $\times \;10^{-70}$), GO:0022625 cytosolic large ribosomal subunit (Bonferroni corrected *p*-value = 9.19 $\times \;10^{-34}$), GO:0005730 nucleolus (Bonferroni corrected *p*-value = 6.91 $\times \;10^{-32}$) (Table 5). The top five GO:MF terms were GO:0044822 poly(A) RNA binding (Bonferroni corrected *p*-value = 6.64 $\times \;10^{-112}$), GO:0003735 structural constituent of ribosome (Bonferroni corrected *p*-value = 2.81 $\times \;10^{-53}$), GO:0003723 RNA binding (Bonferroni corrected *p*-value = 7.61 $\times \;10^{-44}$), GO:0003729 mRNA binding (Bonferroni corrected *p*-value = 8.95 $\times \;10^{-18}$), GO:0098641 cadherin binding involved in cell-cell adhesion (Bonferroni corrected *p*-value = 6.09 $\times \;10^{-17}$) (Table 5). For details about the gene set enrichment (KEGG pathway, GO:BP, GO:CC and GO:MF), four Appendix A, respectively were provided.

## 4. Discussion

In addition, we conducted literature search on the top ten gene markers. Among then ten genes, we obtained nine genes (*Rps21, Slc5a1, Crip1, Rpl15, Rpl3, Rpl27a, Rps3a1, Aldob* and *Rps17*) that had an involvement with some diseases or biological functions, while only one gene (*Khk*) was found to be unknown (novel). For example, the gene *Rps21* was associated with some related biological functions such as artificial nucleic acid molecules [56], exosome and human ribosome biogenesis [57], arterial vasculature [58], a KEGG pathway, Ribosome, and several Gene-Ontologies such as GO:BP: GO:0006412 translation, GO:CC: GO:0005622 intracellular, GO:0022627 cytosolic small ribosomal subunit, GO:MF: GO:0003735 structural constituent of ribosome, GO:0044822 poly(A) RNA binding. Hence, the status of the gene *Rps21* is known. Similarly, the other eight genes, (*Slc5a1, Crip1, Rpl15, Rpl3, Rpl27a, Rps3a1, Aldob* were connected with either some biological functions or KEGG pathway or Gene Ontology or both. For details, see Table 6. However, in case of the gene *Khk*, there was no connection found with any disease or biological function through literature search. Also, there is no single KEGG pathway or GO-term for the gene. Hence, *Khk* gene was a novel maker for the rare intestinal cell types for “Whole Organoid Replicate 1”.

Of note, our study was conducted on a publicly available dataset GEO ID: GSE62270 that was published by Grun et al. [23]. In the dataset, Grun et al. proposed RaceID method and performed an initial analysis with the published dataset. They had selected multiple cell types, such as organoid, Lgr5, Reg4, etc. and then performed analysis on them, whereas we chose only an organoid, “Whole Organoid Replicate 1” that composed of several cell types, and then conducted our analysis using our proposed framework. Our main objective behind our study is to propose a new method to identify cell clusters using multi-objective fuzzy clustering technique for single-cell data. To verify the performance of our method, we used this dataset. The method developed by Grun cell et al. was for rare cell type detection, whereas our method was developed only for cell-cluster identification through multi-objective optimization and fuzzy clustering. As the comparative study, our proposed method and RaceID both used some initial pre-filtering strategy such as cell filtering and gene (feature) filtering. During pre-filtering, we applied a recent robust normalization technique, SCnorm made for single-cell sequencing data, but in RaceID, the total transcript count across each cell was normalized to the median transcript number toward cells. Thereafter, RaceID used k-means clustering algorithm while the number of cluster was determined from “gap statistics” [23], whereas our proposed method used the stronger clustering algorithm, fuzzy c-means in which the number of cluster was determined by a multi-objective optimization technique, TOPSIS on the four cluster validity index measures, PE, PC, MPC, and FSI of which first one was minimization index and remaining three were maximization indices. Since our method focused on optimizing the cell clusters depending upon the quality of clusters, our technique of identifying cell clusters was obviously stronger than the cell-cluster identification approach used in RaceID. On the other hand, RaceID can detect outlier cells (rare cells) from the resultant clusters, whereas we did not do anything in our study to identify outliers (rare cells). In future, we will extend our work to detect rare cells for single-cell RNA-seq data. In case of our analysis, the number of cells was 206 after initial filtering. Here through integrated use of fuzzy clustering and TOPSIS multi-objective optimization technique, we obtained optimal cluster result (TOPSIS optimal score = 0.8584) that comprised of two optimal cell clusters, one containing 115 cells and another with 91 cells. The evaluated scores of the four cluster validity indices, FSI, PE, PC, and MPC for the optimized fuzzy clustering were 0.482, 0.578, 0.607 and 0.215, respectively. In case of the analysis by Grun et al, six cell clusters had been identified through k-means clustering [23].

In technical point of view, our algorithm used fuzzy c-means clustering that is better and more flexible in statistical point of view rather than traditional k-means clustering. In addition, we checked some network evaluating factors such as average scaled connectivity, average maximum adjacency ratio (MAR), density and average Pearson’s correlation. To do so, for our proposed method, we first picked up the set of cells for each evolved cluster, individually and then computed average Pearson’s correlation in cell-pair wise manner for each cluster. We chose the cluster that had highest average correlation. Using the cluster, we determined the above four network evaluating measures, average scaled connectivity = 0.752, average maximum adjacency ratio (MAR) = 0.404, density = 0.295 and average Pearson’s correlation = 0.786. Similarly, for comparative study, we first applied k-means clustering on the data providing the same number of cluster size (=2) as input that was obtained in our proposed method. We then selected the cluster that had highest average correlation, and computed the above four measures, average scaled connectivity = 0.746, average maximum adjacency ratio (MAR) = 0.402, density = 0.294 and average Pearson’s correlation = 0.784. In all these cases, our method provided better scores than those for traditional k-means clustering. Hence, our proposed method provided stronger clustering performance than the existing one.

In addition, we conducted extensive analysis on the two resultant clusters obtained by our method and compared with the 16 clusters (comprising of different rare cell types: goblet, tuft, Paneth and enteroendocrine cells) generated from the random organoid cells of Grun et al. [23] (Appendix A of Grun et al. named as “nature14966-s1.xlsx”). To do so, we first chosen 1240 differentially expressed genes obtained by Limma using the class-label information (cluster #1 vs. cluster #2) of the underlying cells. Thereafter, we checked the fold change of each gene using cell-cluster class labels (each cluster vs. rest), and set the fold change cut off 2 to obtain up-regulated gene markers for the cluster. For cluster #1, we identified 344 up-regulated markers, while for cluster #2, the number of up-regulated markers were 653. After that, we performed intersection of these cluster-markers with the cluster-markers of Grun et al. for the organoid. In the case of our cluster #1 vs. each of the 16 clusters of Grun et al., the number of overlapped gene markers were 13, 61, 45, 9, 11, 4, 67, 18, 49, 25, 79, 29, 18, 52, 63, and 113, respectively. Similarly, for the case of our cluster #2 vs. each of the 16 clusters of Grun et al., the number of overlapped gene markers were 5, 81, 12, 0, 1, 0, 107, 8, 35, 42, 95, 38, 20, 13, 36, and 66, respectively. For detailed information, three Appendix A were provided.

Of note, Limma was generally made for microarray data and later it was extended the model to use it for the RNA-seq data. In details, we first used different pre-filtering techniques and then normalization technique (SCnorm) prior to use Limma for the specific reason. There is some literature where the comparative study among different tools including Limma for scRNA-seq data had been demonstrated [63]. However, currently there are many new tools developed for only scRNA-seq data whose performance is found to be better than Limma to some extent while using scRNA-seq data [63].

## 5. Conclusions

Analysis on the single-cell messenger RNA sequencing data is always a challenging task. In this article, we conducted a comprehensive analysis of detecting the cell clusters and potential gene markers, respectively through a multi-objective optimization-based fuzzy clustering framework for the scRNA-seq gene expression data. In this regard, we initially performed cell filtering as well as gene filtering, and then applied SCnorm normalization. Next, we conducted nine case studies through choosing various cluster sizes (=2, 3, …, 10), and performed fuzzy c-means clustering algorithm individually. From each case study, we measured the scores of the four cluster validity index measures, Partition Entropy, Partition Coefficient, Modified Partition Coefficient, and Fuzzy Silhouette Index. Meanwhile, we fixed these four measures as different objective functions in which first measure was treated as minimization objective, while the rest of these three measures were considered to be maximization objectives, and then applied TOPSIS multi-objective decision-making technique for determining the optimal (best) solution. The case study which contained highest TOPSIS optimal score (top optimal rank), was elected as the best optimal solution. The cluster information of the cells had been detected. After that, we used Limma statistical tool and identified the differentially expressed genes for each resultant cluster with compared to the rest. We applied the count data of “*Whole Organoid Replicate 1*” scRNA-seq dataset for the rare intestinal cell type of *mus musculus*. Using the proposed method, we produced the best optimal solution with TOPSIS optimal score 0.8584 and corresponding optimal clustering result that contained two clusters of which one consisted of 115 cells while the other cluster consisted of 91 cells. The scores of these four cluster validity indices, FSI, PE, PC, and MPC for the optimal fuzzy clustering technique were 0.4818, 0.5784, 0.6073 and 0.2145, respectively. Next, by using Limma, we identified 1240 differentially expressed genes (cluster 1 vs. cluster 2). The top ten gene markers were *Rps21, Slc5a1, Crip1, Rpl15, Rpl3, Rpl27a, Khk, Rps3a1, Aldob* and *Rps17*, respectively among which Khk is novel marker. Gene set enrichment analyses (KEGG pathway and Gene Ontology analyses) had been performed through DAVID online database. Finally, it can say that our proposed framework generated the multi-objective optimized clusters and potential gene markers, respectively that are highly useful for any kind of scRNA-seq data. Of note, the major objective of our proposed method could detect Pareto-optimal cell clusters using multi-objective fuzzy clustering technique for scRNA-seq data. In our method, we had not included any rare cell (outlier) detection strategy. In future work, we will extend our current work to detect rare cells (outliers). 

## Figures and Tables

**Figure 1 genes-10-00611-f001:**
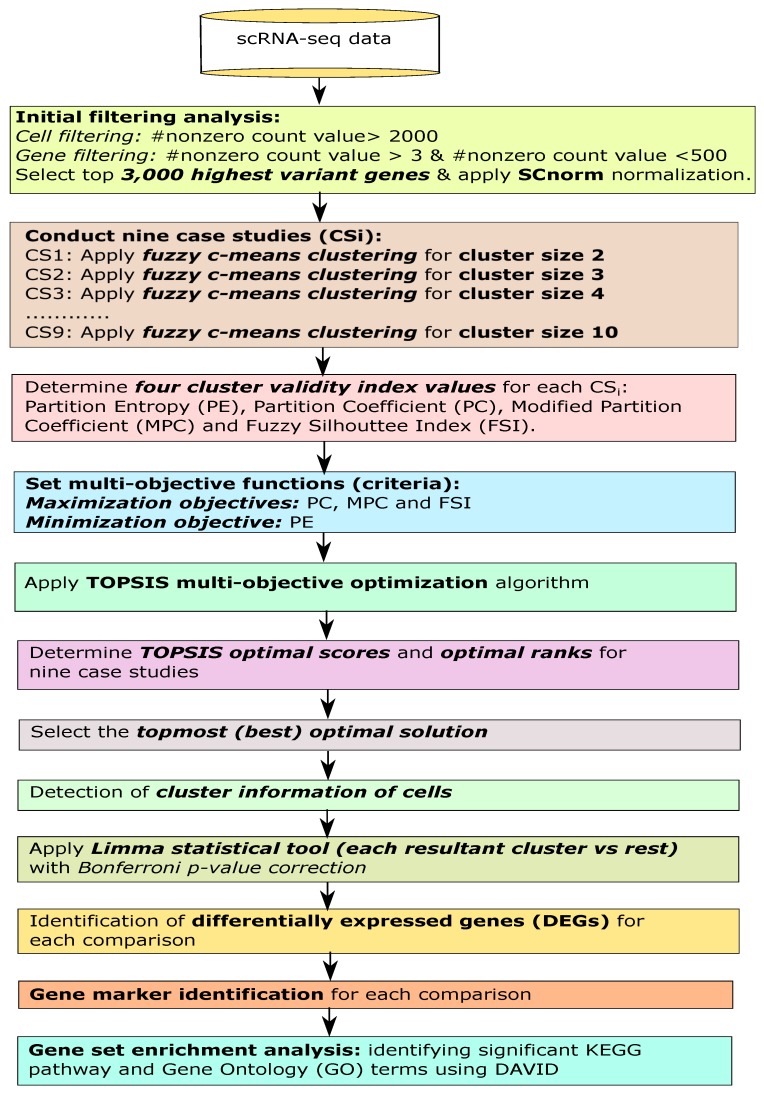
Flowchart of the proposed analysis.

**Figure 2 genes-10-00611-f002:**
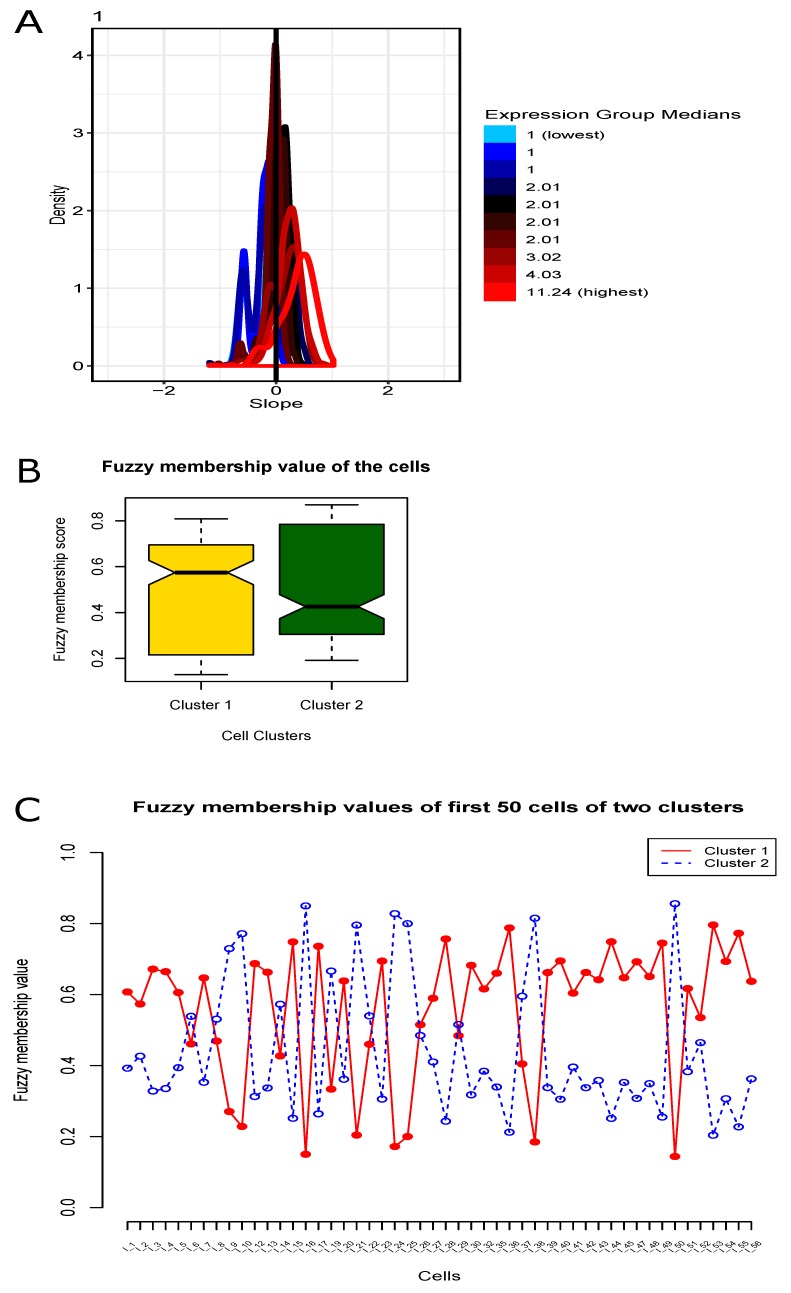
Plots of normalization and fuzzy membership. (**A**) Count-depth relation plot during SCnorm normalization for the scRNA-seq gene expression data. (**B**) Boxplot for the fuzzy membership scores of the cells for the two resultant clusters. (**C**) Fuzzy membership scores of the cells for the two resultant clusters.

**Figure 3 genes-10-00611-f003:**
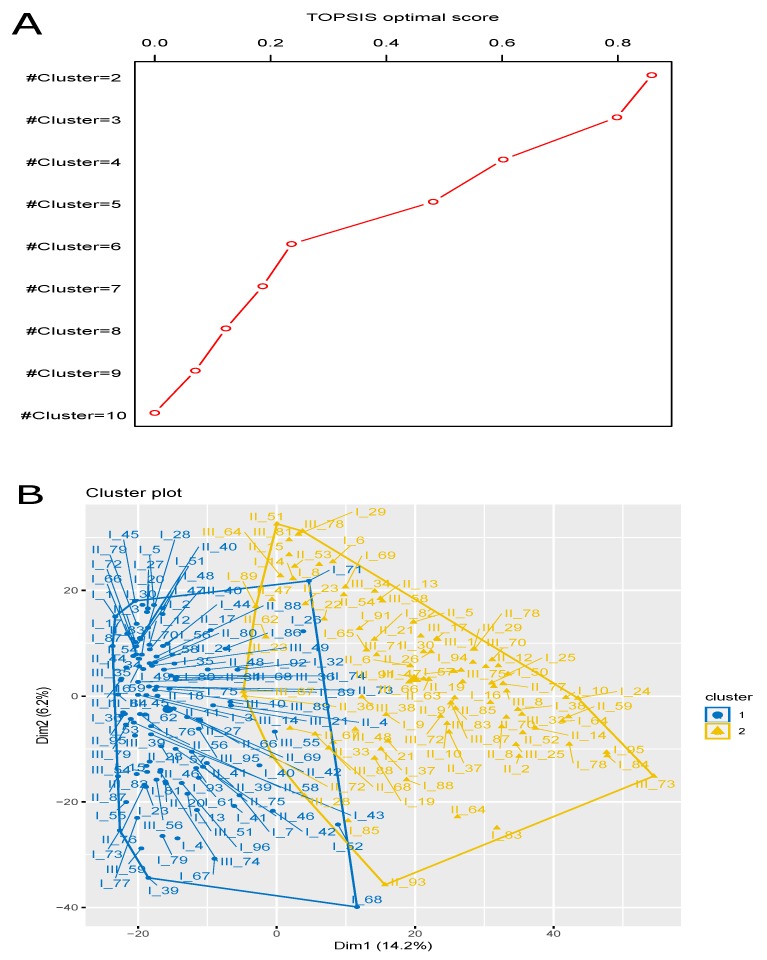
Plots for multi-objective optimization and Principal Component Analysis (PCA). (**A**) Multi-objective optimization (TOPSIS) score for different cluster sizes (nine case studies: cl=2,3,…,10) using Fuzzy c-means clustering. (**B**) The cluster plot (PCA plot) of the optimized fuzzy clustering along with their participating cells.

**Figure 4 genes-10-00611-f004:**
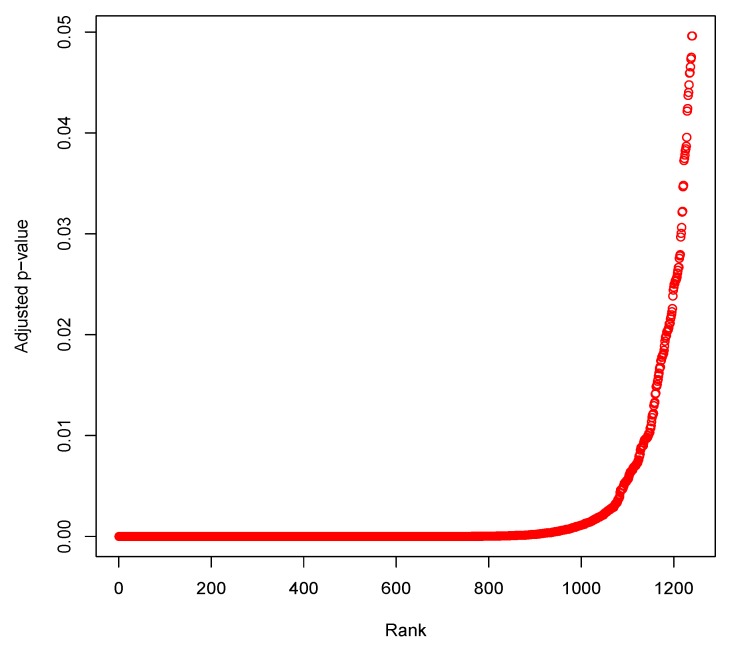
Plot of rankwise Bonferroni adjusted *p*-values for the differentially expressed genes.

**Table 1 genes-10-00611-t001:** The cluster validity scores for the nine case studies from the scRNA-seq gene expression dataset.

Case Study (CS) ID	cl	FSI (↑ ${}^{a}$)	PE (↓ ${}^{b}$)	PC (↑ ${}^{a}$)	MPC (↑ ${}^{a}$)
CS1	2	0.482	0.578	0.607	0.215
CS2	3	0.543	0.886	0.482	0.224
CS3	4	0.588	0.117	0.373	0.164
CS4	5	0.632	0.139	0.304	0.130
CS5	6	0.333	0.157	0.246	0.095
CS6	7	0.364	0.172	0.215	0.085
CS7	8	0.340	0.185	0.190	0.075
CS8	9	0.328	0.197	0.165	0.061
CS9	10	0.267	0.209	0.153	0.059

${}^{a}$ maximization index, ${}^{b}$ minimization index.

**Table 2 genes-10-00611-t002:** TOPSIS optimal scores and optimal ranks for the nine case studies of fuzzy c-means clustering from the scRNA-seq gene expression dataset.

Case Study (CS) ID	cl	TOPSIS Optimal Score	Optimal Rank
CS1	2	0.858	1
CS2	3	0.798	2
CS3	4	0.602	3
CS4	5	0.481	4
CS5	6	0.236	5
CS6	7	0.186	6
CS7	8	0.123	7
CS8	9	0.070	8
CS9	10	0	9

**Table 3 genes-10-00611-t003:** Two resultant clusters and their participating cells after optimized fuzzy clustering (cl=2) from the scRNA-seq gene expression dataset.

Cluster ID	# Cells	Cell IDs
Cluster 1	115	I_1, I_2, I_3, I_4, I_5, I_7, I_12, I_13, I_15, I_17, I_20, I_23, I_26, I_27, I_28, I_30, I_32, I_35, I_36, I_39, I_40, I_41, I_42, I_43, I_44, I_45, I_47, I_48, I_49, I_51, I_52, I_53, I_54, I_55, I_56, I_58, I_59, I_61, I_62, I_66, I_67, I_68, I_70, I_71, I_72, I_73, I_75, I_76, I_77, I_79, I_80, I_81, I_86, I_87, I_92, I_93, I_96, II_1, II_3, II_4, II_11, II_17, II_18, II_20, II_24, II_27, II_28, II_31, II_34, II_39, II_40, II_41, II_42, II_44, II_46, II_48, II_56, II_57, II_58, II_66, II_69, II_73, II_74, II_75, II_76, II_79, II_80, II_83, II_87, II_88, II_89, II_95, III_10, III_14, III_16, III_21, III_35, III_36, III_39, III_40, III_45, III_46, III_49, III_51, III_54, III_55, III_56, III_59, III_68, III_74, III_79, III_82, III_84, III_89, III_95
Cluster 2	91	I_6, I_8, I_9, I_10, I_14, I_16, I_19, I_21, I_22, I_24, I_25, I_29, I_37, I_38, I_50, I_57, I_64, I_65, I_69, I_78, I_82, I_83, I_84, I_85, I_88, I_89, I_91, I_94, I_95, II_2, II_5, II_6, II_8, II_9, II_10, II_12, II_13, II_14, II_15, II_19, II_21, II_23, II_26, II_30, II_33, II_36, II_37, II_47, II_51, II_52, II_53, II_54, II_59, II_62, II_63, II_64, II_67, II_68, II_70, II_72, II_77, II_78, II_85, II_93, III_1, III_8, III_17, III_23, III_25, III_28, III_29, III_33, III_34, III_38, III_47, III_48, III_58, III_64, III_66, III_67, III_70, III_71, III_72, III_73, III_75, III_78, III_81, III_83, III_87, III_88, III_91

**Table 4 genes-10-00611-t004:** Top ten KEGG pathways enriched with the differentially expressed genes.

KEGG Pathway	Count	*p*-Value	Bonferroni *p*-Value	Gene Symbols
mmu03010:Ribosome	94	3.91 $\times \;10^{-62}$	1.08 $\times \;10^{-59}$	*RPL18, RPL17, RPL36A, RPL19, RPL14, RPL13, RPL15, RPLP2, RPS27L, RPL22L1, etc.*
mmu03040:Spliceosome	54	3.72 $\times \;10^{-22}$	1.02 $\times \;10^{-19}$	*SRSF1, LSM6, U2AF2, SNRPD3, LSM7, SNRPD1, SNRPD2, RBM8A, PCBP1, U2AF1, etc.*
mmu01130:Biosynthesis of antibiotics	52	5.87 $\times \;10^{-11}$	1.61 $\times \;10^{-8}$	*SC5D, LDHA, EHHADH, PGAM1, OGDH, CMBL, PKM, IDH3G, PDHA1, CAT, etc.*
mmu03050:Proteasome	21	3.95 $\times \;10^{-10}$	1.09 $\times \;10^{-7}$	*SHFM1, PSMB5, PSMA2, PSMB4, PSMA1, PSMD14, PSMB7, PSMB6, PSMC5, PSMB1, etc.*
mmu01200:Carbon metabolism	33	5.49 $\times \;10^{-9}$	1.51 $\times \;10^{-6}$	*ALDOA, ALDOC, EHHADH, ALDOB, PGAM1, OGDH, GPI1, ACAT1, PKM, TPI1, etc.*
mmu00010:Glycolysis /Gluconeogenesis	23	3.32 $\times \;10^{-8}$	9.13 $\times \;10^{-6}$	*ALDOA, LDHA, ALDOC, HKDC1, ALDOB, FBP1, PGAM1, PFKP, FBP2, GPI1, etc.*
mmu01100:Metabolic pathways	168	7.42 $\times \;10^{-8}$	2.04 $\times \;10^{-5}$	*CYP2C66, CYP2C65, GDA, LDHA, SC5D, CNDP2, EHHADH, CYP2C68, DTYMK, PGAM1, etc.*
mmu00480:Glutathione metabolism	20	1.51 $\times \;10^{-7}$	4.16 $\times \;10^{-5}$	*GSTA1, GSTA2, ODC1, GSTA4, SRM, GGT1, ANPEP, GSTM6, GSTM1, GPX2, etc.*
mmu05204:Chemical carcinogenesis	26	3.58 $\times \;10^{-7}$	9.85 $\times \;10^{-5}$	*CYP2C66, CYP2C65, CYP3A25, CYP2C68, GSTM6, GSTM1, GSTM3, CBR1, GSTM4, ADH1, etc.*
mmu01230:Biosynthesis of amino acids	22	2.30 $\times \;10^{-6}$	6.33 $\times \;10^{-4}$	*ALDOA, SHMT2, MAT2A, ALDOC, ALDOB, PFKP, PGAM1, CPS1, IDH3A, PKM, etc.*

**Table 5 genes-10-00611-t005:** Top five Gene Ontology (GO) terms in each GO domain enriched in differentially expressed genes.

Gene Ontology	Count	*p*-Value	Bonferroni Correction	Gene Symbols
GO:BP ${}^{a}$: GO:0006412 translation	145	3.83 $\times \;10^{-74}$	1.34 $\times \;10^{-70}$	*RPL18, RPL17, RPL36A, RPL19, RPL14, RPL13, RBM3, EIF5, RPL15, EIF5A, etc.*
GO:BP ${}^{a}$: GO:0008380 RNA splicing	67	2.68 $\times \;10^{-26}$	9.38 $\times \;10^{-23}$	*RALY, SRSF1, LSM6, SNRPD3, U2AF2, SNRPD1, SYNCRIP, SNRPD2, YBX1, NONO, etc.*
GO:BP ${}^{a}$: GO:0006397 mRNA processing	75	3.36 $\times \;10^{-24}$	1.18 $\times \;10^{-20}$	*RALY, SRSF1, LSM6, U2AF2, SNRPD3, SNRPD1, SYNCRIP, SNRPD2, YBX1, NONO, etc.*
GO:BP ${}^{a}$: GO:0055114 oxidation-reduction process	101	2.51 $\times \;10^{-17}$	8.79 $\times \;10^{-14}$	*SC5D, LDHA, EHHADH, OGDH, UQCR10, IDH3G, CPOX, PDHA1, HADH, NQO1, etc.*
GO:BP ${}^{a}$: GO:0006413 translational initiation	25	1.07 $\times \;10^{-15}$	3.88 $\times \;10^{-12}$	*ABCE1, EIF5, DENR, EIF1A, LARP1, EIF4B, EIF4G2, EIF3D, EIF3A, EIF3B, etc.*
GO:CC ${}^{b}$: GO:0030529 intracellular ribonucleoprotein complex	151	1.35 $\times \;10^{-99}$	9.24 $\times \;10^{-97}$	*RPL18, MRPL40, RALY, SRP14, RPL17, MRPL42, RPL19, RPL14, RPL13, SNRPD3, etc.*
GO:CC ${}^{b}$: GO:0070062 extracellular exosome	401	3.28 $\times \;10^{-81}$	2.26 $\times \;10^{-78}$	*PRDX5, PRDX2, RPS2, SYNGR2, PTMA, RPS3, SLC1A5, RHOC, TREH, CAT, etc.*
GO:CC ${}^{b}$: GO:0005840 ribosome	101	3.93 $\times \;10^{-73}$	2.70 $\times \;10^{-70}$	*RPL18, MRPL40, RPL17, RPL36A, MRPL42, RPL19, RPL14, RPL13, RPL15, RPLP2, etc.*
GO:CC ${}^{b}$: GO:0022625 cytosolic large ribosomal subunit	50	1.34 $\times \;10^{-36}$	9.19 $\times \;10^{-34}$	*RPL18, RPL17, RPL36A, RPL19, RPL14, RPL13, RPL15, RPLP2, RPL22L1, RPLP0, etc.*
GO:CC ${}^{b}$: GO:0005730 nucleolus	148	1.01 $\times \;10^{-34}$	6.91 $\times \;10^{-32}$	*RPL18, MRPL40, RPL19, LSM6, RBM3, MORF4L2, CBX5, NONO, EBNA1BP2, IMP3, etc.*
GO:MF ${}^{c}$: GO:0044822 poly(A) RNA binding	295	5.08 $\times \;10^{-115}$	6.64 $\times \;10^{-112}$	*RPS25, RPS26, RPS28, PABPC1, RPS20, RPS21, RPS23, HNRNPAB, RPS24, DHX9, etc.*
GO:MF ${}^{c}$: GO:0003735 structural constituent of ribosome	104	2.15 $\times \;10^{-56}$	2.81 $\times \;10^{-53}$	*RPL18, RPL17, RPL36A, RPL19, RPL14, RPL13, RPL15, RPLP2, RPS27L, RPL22L1, etc.*
GO:MF ${}^{c}$: GO:0003723 RNA binding	165	5.82 $\times \;10^{-47}$	7.61 $\times \;10^{-44}$	*RPL18, RALY, SRP14, RPL13, SNRPD3, U2AF2, LSM6, RBM3, LSM7, SNRPD1, etc.*
GO:MF ${}^{c}$: GO:0003729 mRNA binding	46	6.85 $\times \;10^{-21}$	8.95 $\times \;10^{-18}$	*SRSF1, TRA2B, RPL35, RPS2, YBX1, RPS3, HNRNPA3, RPS26, MRPL13, EIF3A, etc.*
GO:MF ${}^{c}$: GO:0098641 cadherin binding involved in cell-cell adhesion	64	4.66 $\times \;10^{-20}$	6.09 $\times \;10^{-17}$	*HSP90AB1, LDHA, RPL14, RPL15, EIF5, PDLIM1, RANGAP1, RPS2, LARP1, BZW2, etc.*

${}^{a}$ Biological Process, ${}^{b}$ Cellular Components, ${}^{c}$ Molecular Function.

**Table 6 genes-10-00611-t006:** Evaluation of the top ten gene markers through literature evidence, KEGG pathway and Gene Ontology analyses.

Gene	Literature Evidence	KEGG Pathway & Gene Ontology Terms	Status
	(Connected with)		
*Rps21*	Biological functions: artificial nucleic acid molecules [56], exosome and human ribosome biogenesis [57], arterial vasculature [58]	**KEGG pathway:** Ribosome (*p*-value = 3.91 $\times \;10^{-62}$), **GO:BP:** GO:0006412 translation (*p*-value = 3.83 $\times \;10^{-74}$), **GO:CC**: GO:0005622 intracellular (*p*-value = 6.8 $\times \;10^{-3}$), GO:0022627 cytosolic small ribosomal subunit (*p*-value = 8.99 $\times \;10^{-24}$), **GO:MF:** GO:0003735 structural constituent of ribosome (*p*-value = 2.15 $\times \;10^{-56}$), GO:0044822 poly(A) RNA binding (*p*-value = 5.08 $\times \;10^{-115}$).	Known
*Slc5a1*	Solute carriers [59]	**KEGG pathway:** Mineral absorption (*p*-value = 1.11 $\times \;10^{-4}$), mmu04973: Carbohydrate digestion and absorption (*p*-value = 9.90 $\times \;10^{-4}$), **GO:BP:** GO:0006810 transport (*p*-value = 2.01 $\times \;10^{-3}$), GO:0001951 intestinal D-glucose absorption (*p*-value = 2.01 $\times \;10^{-2}$), **GO:CC:** GO:0070062 extracellular exosome (*p*-value = 3.28 $\times \;10^{-81}$), **GO:MF:** GO:0015293 symporter activity (*p*-value = 1.93 $\times \;10^{-2}$).	Known
*Crip1*	*Xenopus laevis* embryogenesis [60]	**GO:CC:** GO:0005737 cytoplasm (*p*-value = 6.06 $\times \;10^{-31}$), **GO:MF:** GO:0008301 DNA binding, bending (*p*-value = 2.11 $\times \;10^{-3}$), GO:0042277 peptide binding (*p*-value = 4.27 $\times \;10^{-3}$).	Known
*Rpl15*	Artificial nucleic acid molecules [56]	**KEGG pathway:** Ribosome (pval=3.91 $\times \;10^{-62}$), **GO:BP:** GO:0098609 cell-cell adhesion (*p*-value = 5.06 $\times \;10^{-13}$), GO:0002181 cytoplasmic translation (*p*-value = 2.91 $\times \;10^{-10}$), **GO:CC:** GO:0005739 mitochondrion (*p*-value = 2.25 $\times \;10^{-22}$), GO:0030529 intracellular ribonucleoprotein complex (*p*-value = 1.35 $\times \;10^{-99}$), GO:0070062 extracellular exosome (*p*-value = 3.28 $\times \;10^{-81}$), **GO:MF:** GO:0003735 structural constituent of ribosome (*p*-value = 2.15 $\times \;10^{-56}$).	Known
*Rpl3*	Artificial nucleic acid molecules [56]	**KEGG pathway:** Ribosome (*p*-value = 3.91 $\times \;10^{-62}$), **GO:BP:** GO:0002181 cytoplasmic translation (*p*-value = 2.91 $\times \;10^{-10}$), GO:0042254 ribosome biogenesis (*p*-value = 1.81 $\times \;10^{-09}$), **GO:CC:** GO:0070062 extracellular exosome (*p*-value = 3.28 $\times \;10^{-81}$), GO:0031012 extracellular matrix (*p*-value = 2.06 $\times \;10^{-11}$), GO:0005761 mitochondrial ribosome (*p*-value = 2.56 $\times \;10^{-3}$), **GO:MF:** GO:0003735 structural constituent of ribosome (*p*-value = 2.15 $\times \;10^{-56}$).	Known
*Rpl27a*	Arterial vasculature [58]	**KEGG pathway:** Ribosome (*p*-value = 3.91 $\times \;10^{-62}$), **GO:BP:** GO:0006412 translation (*p*-value = 3.83 $\times \;10^{-74}$), **GO:CC:** GO:0022626 cytosolic ribosome (*p*-value = 5.88 $\times \;10^{-5}$), GO:0022625 cytosolic large ribosomal subunit (*p*-value = 1.34 $\times \;10^{-36}$), **GO:MF:** GO:0003735 structural constituent of ribosome (*p*-value = 2.15 $\times \;10^{-56}$).	Known
*Rps3a1*	-	**KEGG pathway:** Ribosome (*p*-value = 3.91 $\times \;10^{-62}$), **GO:BP:** GO:0006412 translation (*p*-value = 3.83 $\times \;10^{-74}$), GO:0043066 negative regulation of apoptotic process (*p*-value = 5.39 $\times \;10^{-5}$), **GO:CC:** GO:0070062 extracellular exosome (*p*-value = 3.28 $\times \;10^{-81}$), GO:0022627 cytosolic small ribosomal subunit (*p*-value = 8.99 $\times \;10^{-24}$), GO:0030529 intracellular ribonucleoprotein complex (*p*-value = 1.35 $\times \;10^{-99}$), **GO:MF:** GO:0044822 poly(A) RNA binding (*p*-value = 5.08 $\times \;10^{-115}$).	Known
*Rps17*	Different biological functions: proteomic analysis [56,57,58,61]	**KEGG pathway:** Ribosome (*p*-value = 3.91 $\times \;10^{-62}$), **GO:BP:** GO:0000028 ribosomal small subunit assembly (*p*-value = 1.17 $\times \;10^{-7}$), **GO:CC:** GO:0070062 extracellular exosome (*p*-value = 3.28 $\times \;10^{-81}$), GO:0005739 mitochondrion (*p*-value = 2.25 $\times \;10^{-22}$), GO:0031012 extracellular matrix (*p*-value = 2.06 $\times \;10^{-11}$), **GO:MF:** GO:0003735 structural constituent of ribosome (*p*-value = 2.15 $\times \;10^{-56}$).	Known
*Aldob*	Hepatocellular cellular carcinoma [62].	-	Known
*Khk*	-	-	**Novel**

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
