# Peer review of "Multi-Objective Optimized Fuzzy Clustering for Detecting Cell Clusters from Single-Cell Expression Profiles"

_genes, 2019, doi:10.3390/genes10080611_

Round 1

Reviewer 1 Report

In this manuscript authors describe a method for clustering single-cell RNA-seq data using Fuzzy c-means clustering. But the manuscript is not well written (both in content and to some degree, language), there are missing figures and a lot of redundancy within the text. Therefore, it is difficult to judge the novelty and usefulness of their method over existing methods for single-cell RNA-seq clustering.

1)   The authors should provide a comparison of clustering between their method and other popular method of choice eg:- K-means clustering . For instance, here they get 2 clusters from the organoid samples. How many clusters would K-means identify? What do the two clusters identified here mean biologically, are they representing two major cell types found in the organoid?

2)   Please describe count-depth relation plot in Fig 2 and what is the reason for showing it.

3)   The figure reference indicates Fig 2(A), Fig 2(B) and Fig 2 (C), but figure 2 doesn't have these panels. Similarly, there is no Fig 3(B). 

4)   Line 86-88 – the authors mention previous studies have used fuzzy clustering in single cell RNA-seq data but provide no references.

5)   Lines 129-134 which describes the initial data filtering steps are repeated verbatim in lines 217-222. Please describe this only once either in the result section or the method section. Similarly, the entire section 3.6 is basically a repetition of Tables 4 and 5 GO analysis. A large part of discussion is also a mere reporting of gene ontology (GO) analysis output which is not expected in a discussion section.

6)   “Whole Organoid Replicate 1” is not a cell type (Line 349). It’s an organoid composed of several cell types.

Author Response

Reviewer #1:

In this manuscript authors describe a method for clustering single-cell RNAseq data using Fuzzy c-means clustering. But the manuscript is not well written (both in content and to some degree, language), there are missing figures and a lot of redundancy within the text. Therefore, it is difficult to judge the novelty and usefulness of their method over existing methods for single-cell RNA-seq clustering.

Response: We thank the reviewer #1 for carefully reading our manuscript and providing us valuable suggestion. We are sorry that we did not write the manuscript clearly and concisely so that the reviewers/readers can judge on the novelty and usefulness of the method over existing methods for single-cell RNA-seq (scRNA-seq) clustering. In revision, we tried our best to enhance the writing, reduce the redundancy, and clarify the novelty and performance of our method.

In literature, there are many articles about the identification of gene signature [25–28] or, biomarker [29–31] or, gene module [26,32,33] or, transcriptome analysis [34,35] for microarray/RNA-seq data and the integration of multi-omics data [36–40]. Among them, there are a few papers using fuzzy clustering for microarray/RNA-seq data [41, 42]. However, there has been no study yet using multi-objective optimization and fuzzy clustering together on single cell RNA-seq data. Therefore, we thought that our work is the first to introduce a computational framework to identify cell clusters using multi-objective optimization and fuzzy clustering together on scRNA-seq data. Our method has several advantages in the application of fuzzy clustering and TOPSIS multi-objective optimization strategy, as summarized below.

(i) Fuzzy C-Means (FCM) is a kind of clustering strategy in which each sample point belonging to the cluster is characterized by its membership function. In general, FCM tries to maintain the membership matrix of the input dataset that has been updated on every iteration by estimating the associated weight of every sample point to evaluate its degree of membership. The summation of every sample point towards all the clusters is unity. The main benefits of this strategy to scRNA-seq data include its capability to form clusters of the overlapped sample points and the results satisfy the property of convergence. The potential limitations of the cluster validity are that the prior necessity of c value is required for the quality clustering outcomes and outliers might be assigned to the similar membership value in every cluster. These limitations make it less desirable for using any kind of gene expression data.

(ii) TOPSIS method is used to identify the set of multi-objective optimized clusters. In the scRNA-seq data, the number of cell clusters varies among data sets. In this study, we attempted to identify the best set of multi-objective optimized clusters as measured by the quality of clustering, i.e., different clustering validity index measures.

(iii) We performed a comparative analysis of our proposed method with the existing k-means clustering method. The comparative analysis indicated that our method outperformed over k-means clustering method.

We mentioned this in the second paragraph, page 3. In addition to that, we also performed text writing enhancement in many places, added more detailed information as per reviewer’s suggestion and added the missing sub-figure labels in Figures 2 and 3. This helps to read the figures easily.

We also would like to note that this work was selected as an oral presentation in the International Conference on Intelligent Biology and Medicine held on June 9-11, 2019. The audience had very positive feedback on this work during the presentation. They praised our idea and novel approaches to the scRNA-seq data.

Comment #1: The authors should provide a comparison of clustering between their method and other popular method of choice eg:- K-means clustering . For instance, here they get 2 clusters from the organoid samples. How many clusters would K-means identify? What do the two clusters identified here mean biologically, are they representing two major cell types found in the organoid?

Response: We have already compared our proposed multi-objective clustering method with k-means clustering, as in our previous revision to respond to the comment from an ICIBM peer reviewer. In our method, we obtained cluster size =2 as an optimal cluster size. Since in k-means clustering users have to provide the desired (user-defined) cluster size, we provided the evaluated cluster size (=2) as input in k-means for comparison. Actually, we wanted to observe the effect of the quality of clusters while both the methods have same cluster size. Only difference is that we evaluated the optimal cluster size through multi-objective optimization and then used fuzzy clustering, while in k-means, users have to provide the cluster size directly as input. However, we then evaluated four measures, average scaled connectivity, average maximum adjacency ratio (MAR), density and average Pearson’s correlation. For our proposed method, the values of the above four network evaluating measures were mentioned as follows: average scaled connectivity = 0.752, average maximum adjacency ratio (MAR) = 0.404, density = 0.295 and average Pearson’s correlation = 0.786, whereas for k-means, there were 0.746, 0.402, 0.294 and 0.784, respectively. We mentioned these in Discussion section, page 16-18, lines 362-376.

As suggested by the reviewer #1, we conducted extensive analysis on the two resultant clusters obtained by our method and compared with the sixteen clusters (comprising different rare cell types: goblet, tuft, Paneth and enteroendocrine cells) generated from the random organoid cells of Grun et al. [23] (Supplementary Table 1 in Grun et al, the file name “nature14966-s1.xlsx”). To do so, we first chose 1,240 differentially expressed genes obtained by Limma using the class-label information (cluster #1 vs cluster #2) of the underlying cells. Thereafter, we checked the fold change of each gene using cell-cluster class labels (each cluster vs the rest), and set the fold change cut off 2 to obtain up-regulated gene markers for the cluster. For cluster #1, we identified 344 up-regulated markers, while for cluster #2, the number of up-regulated markers were 653. After that, we performed intersection of these cluster-markers with the cluster-markers of Grun et al for the organoid. In the case of our cluster #1 vs each of the sixteen clusters of Grun et al., the number of overlapped gene-markers were 13, 61, 45, 9, 11, 4, 67, 18, 49, 25, 79, 29, 18, 52, 63 and 113, respectively. Similarly, for the case of our cluster #2 vs each of the sixteen clusters of Grun et al., the number of overlapped gene-markers were 5, 81, 12, 0, 1, 0, 107, 8, 35, 42, 95, 38, 20, 13, 36 and 66, respectively. We mentioned these in Discussion section, page 18, lines 377-391. For details, please see supplementary files Tables S5-S7 in our revised manuscript.

Comment #2: Please describe count-depth relation plot in Fig 2 and what is the reason for showing it.

Response: We thank the reviewer for raising this question. Here is the rationale of Figure 2. Prior to normalize using SCnorm normalization [44], the relationship between the expression counts and the corresponding sequencing depth (named as the count-depth relationship or slope) for the experimental data should be verified.

SCnorm normalizes across the cells for eliminating the effect of the sequencing depth on the counts. The genes were initially partitioned into 10 equally sized groups depending upon their non-zero median expression. In SCnorm, only those genes that had at least 10 non-zero expression values were selected by default. SCnorm initiated at the value of the parameter K being equal to 1; this helped to normalize the data with the assumption that all the genes had to be normalized in a single group. The sufficiency of the score K = 1 was estimated through determining the normalized count-depth relationship. To do so, all the genes were divided into 10 groups depending on their corresponding non-zero unnormalized median expression scores (considering equal group size) and evaluated the mode for each corresponding group. Whenever all 10 modes were within 0.1 of zero, the value K = 1 would be sufficient. While any of those modes was less than -0.1 or greater than 0.1, the SCnorm method attempted to normalize by considering K = 2 and repeated the group-wise normalization along with the corresponding estimation. It would continue until all those modes were within 0.1 of zero. Moreover, SCnorm method initially attempted to fit the corresponding model for the value K = 1, and then subsequently increased the value of K until an approximate satisfactory stopping point could reach. In our study, the value of K was evaluated as 4 (the four groups of slopes colored as cyan, blue, deep brown and red ordered by expression median score from low to high) while all those 10 slope densities had the absolute slope mode < 0.1 (default cutoff). However, the normalization of the highly expressed genes is expected to be good, while the lowly expressed (and moderately expressed) genes might be over-normalized; in that case, they generated negative slopes. Please see Figure 2(A) for details.

As the gene expression generally increased proportionally when increasing the sequencing depth, the count-depth relationships were required to evaluate near to the value 1 for all the genes. In general, for single cell data, those relationships varied across the genes. However, after the SCnorm normalization, the count-depth relationship could be evaluated on the normalized data profile where the slopes near zero signified successful normalization.

We mentioned the necessity and description of count-depth relationship plot in Section 2.1, the second paragraph, page 5, lines 143-164 and Section 3.2, page 9, lines 248-254.

Comment #3: The figure reference indicates Fig 2(A), Fig 2(B) and Fig 2 (C), but figure 2 doesn't have these panels. Similarly, there is no Fig 3(B).

Response: We were very sorry for those missing subpanels in the figures. We included in the revised manuscript. See page 10 for Fig. 2 and page 12 for Fig. 3 in the revised manuscript.

Comment #4: Line 86-88 – the authors mention previous studies have used fuzzy clustering in single cell RNA-seq data but provide no references.

Response: Thank you so much for this comment. We have revised it as follow (revised manuscript, page 3, lines 86-92).

“In literature, there are many articles about the identification of gene signature [25–28] or, biomarker [29–31] or, gene module [26,32,33] or, transcriptome analysis [34,35] for microarray/RNA-seq data and the integration of multi-omics data [36–40]. Among them, there are a few studies conducted using fuzzy clustering for microarray/RNA-seq data [41], [42]. However, there has been no study yet using multi-objective optimization and fuzzy clustering together on single cell RNA-seq (scRNA-seq) data yet. Here, we proposed a new computational method to identify cell clusters using multi-objective optimization and fuzzy clustering together on scRNA-seq data.”

Comment #5: Lines 129-134 which describes the initial data filtering steps are repeated verbatim in lines 217-222. Please describe this only once either in the result section or the method section. Similarly, the entire section 3.6 is basically a repetition of Tables 4 and 5 GO analysis. A large part of discussion is also a mere reporting of gene ontology (GO) analysis output which is not expected in a discussion section.

Response: We thank the reviewer for this valuable comment. As suggested, we eliminated the redundant text from the “filtering” subsection in the revised manuscript. In the Materials and Methods section, we described the filtering procedure that we developed. In the Results section, we only mentioned the results of cell filtering and gene filtering for the specific dataset used. Please see Section 2.1, page 4-5, lines 132-140 and Section 3.2, page 9, lines 246-248.

We also eliminated the redundant information of pathway/GO analysis form Discussion. We made it short and concise. Accordingly, we added Table 6 to provide all the detail of the literature search, KEGG pathway and Gene Ontology terms associated with each gene marker. Please see the first paragraph of Section “Discussion”, page 16, lines 319-332 and Table 6 in page 17.

Comment #6: “Whole Organoid Replicate 1” is not a cell type (Line 349). It’s an organoid composed of several cell types.

Response: We thank this reviewer for pointing out. We made the change in the revised version. See page 16, lines 336-337 (“Discussion” section).

Reviewer 2 Report

In this manuscript, the authors proposed a multi-objective decision-making  model for detecting cell clusters from scRNA-seq data. By integrating four clustering validity index measures (i.e., Partition Entropy, Partition Coefficient, Modified Partition Coefficient and Fuzzy Silhouette Index), the authors applied a multi-objective decision making technique, referred to as TOPSIS, to obtain the final optimal results of clustering. Furthermore, the differentially expressed genes and gene makers with important biological functions were identified based on the clustering results. Overall, the proposed method of this manuscript is of general interest and will likely lead to better performance and results in cell clustering and scRNA-seq data analyses. I recommend "revise and accept". Below are several concerns about the current version of this manuscript.

For a nontrivial multi-objective optimization problem, there is no single solution that optimizes each objective simultaneously. In that case, the objective function are said to be conflicting, and there exists a number of Pareto Optimal Solutions (POSs). In general, we can select the reasonable solution in POSs based on the actual situation. However, the proposed method in this manuscript can obtain a unique solution, which is chose with the highest TOPSIS score. I am not sure if this is the optimal solution. In addition, how does the author determine the weights of the four objective functions?

In Section 2.2, the authors applied fuzzy c-means clustering for the initial number of clusters. Why the authors set a maximum cluster number of 10?

Many thresholds were selected for filtering results, such as the sum score greater than 2000 for cell filtering and the sum of all non-zero values greater than 3 and less than 500. It would be helpful if the authors provide explanation of how these thresholds were determined.

Author Response

Reviewer #2:

In this manuscript, the authors proposed a multi-objective decision-making model for detecting cell clusters from scRNA-seq data. By integrating four clustering validity index measures (i.e., Partition Entropy, Partition Coefficient, Modified Partition Coefficient and Fuzzy Silhouette Index), the authors applied a multi-objective decision making technique, referred to as TOPSIS, to obtain the final optimal results of clustering. Furthermore, the differentially expressed genes and gene makers with important biological functions were identified based on the clustering results. Overall, the proposed method of this manuscript is of general interest and will likely lead to better performance and results in cell clustering and scRNA-seq data analyses. I recommend "revise and accept". Below are several concerns about the current version of this manuscript.

Response: We thank the reviewer #2 for carefully reading our manuscript and positive comment on our work.

Comment #1: For a nontrivial multi-objective optimization problem, there is no single solution that optimizes each objective simultaneously. In that case, the objective function are said to be conflicting, and there exists a number of Pareto Optimal Solutions (POSs). In general, we can select the reasonable solution in POSs based on the actual situation. However, the proposed method in this manuscript can obtain a unique solution, which is chose with the highest TOPSIS score. I am not sure if this is the optimal solution. In addition, how does the author determine the weights of the four objective functions?

Response: Thank you so much for this interesting comment.  In our work, we used TOPSIS method which is basically a multi-criterion decision analysis method. In this method, the relative closeness (RC) score was finally evaluated from the scores of multiple criteria (multiple objectives) used in our study through determining positive ideal solution and negative ideal solution. Among the objectives, three (Partition Coefficient, Modified Partition Coefficient and Fuzzy Silhouette Index) were maximization index, whereas remaining objective (Partition Entropy) was minimization index. Therefore, the TOPSIS optimal score (=RC) depends upon difference between those scores including specific terms and conditions (maximization and minimization conditions). Hence, TOPSIS does not produce any biased result. It generated the Pareto Optimal result according to the objective functions for the corresponding data. Of note, if the data is changed, the Pareto Optimal solution might be different.

In our cases, we provided equal weight (preference) to each of the four objective functions since these four objectives are equally important for evaluating the quality of clustering. We mentioned this information in Section 2.4, page 8, lines 211-212.

Comment #2: In Section 2.2, the authors applied fuzzy c-means clustering for the initial number of clusters. Why the authors set a maximum cluster number of 10?

Response: We thank the reviewer for this critical comment. In TOPSIS multi-objective optimization method, the relative closeness (RC) score was finally estimated from the scores of multiple criteria (multiple objectives) used here through determining positive ideal solution and negative ideal solution. As shown in Figure 3 (A), the TOPSIS optimal score decreases every time when the number of clusters decreases. In addition, in the experimental data, there was small number of cells (only 206 cells) in the single organoid that we chose before clustering. Therefore, we checked the first nine cases (#cluster = 2 to 10) because those clusters had good scores to evaluate. The case study that had highest optimal score, was finally considered as the final outcome of clustering, where other case studies were simply neglected. In addition, since the cell size in the data was small and the curve of optimal score was going down every time when increasing the cluster size, there was no need to check for the cases having cluster size >10. We mentioned this information in Section 3.4 page 11 lines 275-282.

Comment #3: Many thresholds were selected for filtering results, such as the sum score greater than 2000 for cell filtering and the sum of all non-zero values greater than 3 and less than 500. It would be helpful if the authors provide explanation of how these thresholds were determined.

Response: Thank you so much for the comment. We checked many papers regarding this filtering cutoff procedure such as Grun et al. [23], Talwar et al. [43], etc., and realized that filtering criteria are not fixed anymore. Hence, we chose the standard cutoffs for cell filtering as well as gene filtering. However, after filtering, we selected the 3000 top variant genes for the next stage of analysis. We mentioned this information in Sub-section 2.1 pages 4-5, lines 132-140.

Round 2

Reviewer 1 Report

The authors have addressed all my comments and included the figures that were missing in the previous version. One minor suggestion I have is that, several figures appear slightly distorted. The figure presentation should be improved before publication.